# On Russell's 1927 Book *The Analysis of Matter*

**Said Mikki** [1,2]

1   Zhejiang University/University of Illinois at Urbana-Campaign (ZJU-UIUC) Institute, Haining 314400, China; said.mikki@rmc.ca
2   Royal Military College of Canada, Kingston, ON K7K 7B4, Canada

**Abstract:** The goal of this article is to bring into wider attention the often neglected important work by Bertrand Russell on the philosophy of nature and the foundations of physics, published in the year 1927. It is suggested that the idea of what could be named *Russell space*, introduced in Part III of that book, may be viewed as more fundamental than many other types of spaces since the highly abstract nature of the topological ordinal space proposed by Russell there would incorporate into its very fabric the emergent nature of spacetime by deploying *event assemblages*, but not spacetime or particles, as the fundamental building blocks of the world. We also point out the curious historical fact that the book *The Analysis of Matter* can be chronologically considered the earliest book-length generic attempt to reflect on the relation between quantum mechanics, just emerging by that time, and general relativity.

**Keywords:** Bertrand Russell; event ontology; history of philosophy; foundations of physics



## 1. Introduction

In 1927, Bertrand Russell published a very remarkable book entitled *The Analysis of Matter* [1], which is based on a series of lectures he delivered sometime earlier. My goal here is not turning out another review of this nearly one-hundred-year-old book. Instead, my purpose is to provide some general remarks on the overall philosophical scope of such quite unusual work in the history of ideas, especially in regard to its possible connection with fundamental research on the nature of space [2], in addition to the still ongoing problem of finding a general working theory of quantum gravity [3–6]. Indeed, the completion of modern quantum mechanics in 1927 coincided with the publication of this work. As is well known, Russell had always been very informed about the latest development in physics and mathematics [7,8], and, in the book, he cites the just published fundamental papers by Heisenberg, Dirac, Weyl, and other founders of modern physics, which had been being turned out during the extraordinary productive five-year period 1923–1927 that eventually shaped our understanding of physics and nature [9]. However, Russell's acute awareness of the radical changes taking place in science during that time, most importantly, the fundamentally *nonclassical* nature of the microscopic world of quantum mechanics, did not deter him from pursuing the very difficult task of building a generalized theoretical framework for space that could be used for various applications, for example, shedding light on the relation between perception and reality, the microscopic world and the nature of spacetime, or between gravity and the quantum world. He attempted to do so using both *philosophical* means (Part I and II), but, more interestingly, through *mathematical philosophy* (Part III). His background as probably the most important mathematical philosopher of the twentieth century [10,11] did eventually help in making the formulation more comprehensive than earlier and even later works.

My main opinion in this short article can be summarized by the following two observations:

1.   Russell had embraced, in Part I and Part II of his book, a somehow traditional "Russellian" style of doing the philosophy of science, where emphasis is usually laid

on the relation between nature and perception (British empiricism) and the associated metaphysics, e.g., in the spirit of Whitehead's earlier work [12].

2.　The real contribution of Russell's book, however, is in Part III, which is often neglected by mathematicians and philosophers of science. Here, his tone had shifted noticeably from traditional philosophy of science toward fundamental ontology, in the style of some of his other books such as [10,11,13–15].

I deal with the first observation in Section 2 below, where the metaphysical and epistemological aspects of Russell's presentation are revisited (very briefly.) On the other hand, the more radical development pertaining to fundamental ontology are dealt with (admittedly only at the high level) in Section 3. Detailed mathematical definitions and rigorous constructions of Russell space are not provided here. Instead, the interested reader is strongly advised to browse through Part III of Russell's book, which can be read independently of Parts I and II. Some remarks on the historical interpretation of the evolution of quantum gravity and Russell are provided in Section 4, where the purpose is to clarify our position with regard to how contemporary theories of quantum gravity may be compared with past figures such as Leibniz, Riemann, or Russell. Russell's position is briefly criticized in Section 5, where it is suggested that we may need to go beyond Russell (and Whitehead) in terms of the manner by which the event is defined. Finally, I end up with conclusions.

## 2. Metaphysics vs. Epistemology and the Shift toward Fundamental Ontology in Russell's Account

Part I may be viewed as a philosophical introduction to the latest developments in gravitation and quantum theory. For the classical theory (general relativity), he draws on the works of Weyl [16] and Eddington [17,18] to reformulate the problem in the most abstract and far-reaching form possible (gravity as a gauge field theory, though, naturally enough, the present modern form [19] is still not there yet). Within the entire content of Part I, the most remarkable chapter is the one entitled *Measurement*, which, in spite of being not very conclusive, succeeded in providing some of the most penetrating remarks on the general abstract formulation of the problem of measurement in theoretical physics I am aware of. In this particular location, one can feel resonances with the third volume of *Principia Mathematica* [15], i.e., the ontology of magnitudes, which reaches back to certain ideas in *The Principles of Mathematics* [10]. However, I will say nothing more on measurement in the remainder of this paper.

Part II deals mainly with epistemological issues and is probably the least remarkable in the entire book. Here, the now classic attention often paid by British philosophers to the problem of the relation between sense perception and sense data, on the one hand [20], and the construction of theories of nature, on the other [21], is highlighted and developed in a series of short chapters typical of Russell. The future (and last important) 1948 book by Russell, *Human Knowledge* [22], would somehow attempt a grand synthesis of these two dimensions of the philosophy of nature in which epistemological considerations related to the problem of perception are linked to the deductive and inductive (probabilistic) foundations of certainty and belief in theory construction processes. While, in my opinion, there has not been much attention paid to Russell's 1927 book within the secondary literature in general, the notable few exceptions tend to concentrate on the global epistemological aspects of the work [23].

If Part I can be loosely described as a "metaphysical account" of classical and quantum physics, while Part II is mainly concerned with the epistemological foundations behind our attitude toward nature, then Part III qualifies as the "ontological turn" of Russell's thought, where the philosopher of nature now strives to re-calibrate his resources in order to take up the fundamental ontology of the world at the very basic and abstract level of his best (earlier) works [10,11]. The various chapters in this part deal with a diverse array of highly technical subjects, probably more than in any other book by Russell (with the exception of *Principia* and the *Principles of Mathematics*). Building on the emerging subject

of general (set-theoretic) topology, especially the theories of Cantor [24], Urysohn [25], and Hausdorff [26], Russell's *very* ambitious aim is to propose the most general concept of "space" possible, with an eye on several applications in epistemology, foundations of physics, and even possibly "quantum gravity" as understood by Russell's time.

There are several notable technical features in the Russell's program of Part III that will be taken up again in Section 3. For now, I would like to mention the all-important (and currently quite popular) theme of the *emergence* of spacetime. This is in fact a theoretical motif that was very well alive not only in the mind of Russell himself, but also Whitehead [12,27–29], Jakob von Uexküll [30], and, before them, even Mach [31] and William James [32] through their respective versions of neutral monism. The reason why Russell added the very technical Part III is that he wanted to defend a version of ontological monism that was dominant in his thinking at the time. The idea is that nature in its essence is both *abstract* and *material*.

I propose to call this philosophical program *abstract materialism*, which is a rather specific view of nature that should be distinguished from various other versions such as the non-dualist ontologies of Spinoza [33], Leibniz [34], Schelling [35], Bergson [36–38], James [32], and Mach [31]. While all these ontologies (and Russell's) can be united in being a revolt against the dualism of Descartes and Kant (and possibly Hegel), they differ from the Russellian attitude by the degree in which their major orientation derives from the concept of *structure* and its various derivative thematic points of view. As a matter of fact, the fundamental concept of *structure* in physics, coupled with a fully fledged structure-centric approach to the philosophy of nature (nature *is* structure), a position already defended by Eddington [39], was taken up again by Russell's book but now implemented using the very sophisticated topological toolkits available from earlier Russellian forays into mathematics, logic, language, and epistemology [10,11,21,40,41]. One of the results obtained from such ambitious endeavour is something totally new and unprecedented: It is what I intend to dub *Russell space*, while proposing the opinion that it constitutes possibly one of the most interesting general topological spaces in the literature.

## 3. The Concept of Russell Space

As I did with regard to the remarks on Parts I and II, no detailed examination of the contents of Part III will be given here. My objective is to reconstruct an interpretation of those particular elements in Russell's overall program that can be located in the outstanding (but rarely read) third division of his book. Readers interested in more mathematical details should directly read Russell's original text, but they may benefit by first refreshing their memory about ordinal numbers and order relations methods, topics often discussed in connection with Cantor's point-set topology [24] in texts devoted to abstract set theory, e.g., see [26,42,43]. The topological ideas themselves were somehow stimulated by Russell's early masterpiece [10], which was widely read by all mathematicians, including Hilbert and Poincare [7]; e.g., see how the book [10] inspired the construction of Frechet space, one of the earliest abstract spaces in modern mathematics [25], which lead to other development such as Banach spaces for instance. However, it must be noted that Russell space is based on *events*, not sets of points. This presents a particular difficulty in regard to how to define operations on events without reference to a "total set" (the set of all events is not a point set, nor the event itself). For example, should interaction between events be considered a set-theoretic intersection? To avoid these difficulties, Russell used the predicate calculus that he himself helped create in the early years of the twentieth century in order to axiomatically define event-event coupling without invoking sets of points (this is important since the whole issue, as will be seen below, is to derive spacetime points themselves from events). Our message here is that the culmination of all such factors (and others not discussed here) had contributed to making the mathematics used by Russell for constructing his Russell space concept unfamiliar, and even possibly "strange-looking" with respect to the sensibilities of our present-day style of doing mathematical physics and mathematical philosophy. An alternative, then, following the reading of Russell's text, would be the

invention of equivalent or closely related mathematical formalisms, based on contemporary mathematical physics and mathematics, capable of either emulating Russell space or even "transcending" it by moving into a higher realm of mathematical representation. Clearly this is a work that belongs to future research.

While Parts I and II of Russell's book have already drawn some critical reviews and examinations in the secondary literature, it remains true that, in the main, there has been a tendency to focus on the later book *Human Knowledge* [22], first published in 1948, which contains some of the ideas developed in *The Analysis of Matter*, Part III. However, careful reading of the two books reveals essential differences. *Human Knowledge*, as the title explicitly says, focuses on the foundations of our knowledge of nature, i.e., it is essentially a work on the epistemology of the natural sciences. Since Russell is no ordinary epistemologist—his worldview already overladen with a heavy emphasis on metaphysics and ontology—it is only natural that his treatment of the subject of the foundations of scientific knowledge in the 1948 text will rework some of the technical constructions presented in Part III of the 1927 book. Nevertheless, the main emphasis of the epistemological approach is not on providing a general ontological matrix for the genesis of being, as I suggest *was* in fact his intention in the earlier work *The Analysis of Matter*, but rather the narrower goal of integrating Russell's ontological theory of causality with the probabilistic inductive foundations of scientific knowledge in general, and the physical sciences in particular.

A need arises then to establish a more focused interpretive stand with respect to Part III of the 1927 book, emphasizing its uniqueness and distinctiveness not only within the Russell Universe, but in the history of fundamental physics. There are two main objectives motivating my interpretation:

1.  Providing a very rough outline for a sketch of what I proposed, above, to call *Russell space* without burdening my presentation with full mathematical details (these can be found in Russell's book.)
2.  I seek to suggest a curious parallelism between Russell's book and some "contemporary new ideas" on the emergence of spacetime in nature, especially within the framework of present-day quantum gravity, despite the fact that the latter is extremely different from Russell's at the lower-level side of the technical content.[1]

These two objectives will be briefly addressed in order. More detailed examination will be given in a future text.

What is the main idea of Russell space? Is this nothing but another example of one of those contemporaneous or future (with respect to Russell's time) "famous spaces" now populating, even sometimes overcrowding, mathematical physics, such as Frechet space, Hilbert space, Banach space, Gelfand space, and so on? The very idea of an abstract mathematical space was relatively new by Russell's book's time, having been introduced into the field only a quarter of century earlier by several authors, chief among them is Russell himself in his 1900–1903 magnum opus *The Principles of Mathematics* [10,40]. But the germs of a concept do not contain the entire story; for, throughout the intervening years, Russell had been integrating several key ideas into the basic structure of general abstract mathematical space, notably the principle of *causality*[2] and the ontology of the *event* [21].

Russell space is a supra-Cantorian generative matrix of structures. That is, it is not a concrete or a specific space such as the complex plane or Hilbert space, but a very abstract *reconfigurable* dynamic "metastructure" capable of "rewiring" itself in order to produce new structures. A Russell space, then, is a *context-driven ontological framework for concretization*. A given space *becomes* concrete when various, already individuated, fundamental elements (events) interact with each other in accordance to the Russellian principle of generalized causality, which is essentially a topological order relation with

---

1    Cf. Section 4.

2    Russell's concept of causality is not identical to relativistic causality. The former is more general and is topological in nature. It has many interpretations in the literature. Reichenbach's interpretation of Russell's idea developed in [44] is particularly interesting.

specific characteristics. The interaction of several nexuses of events may lead to the production of concrete/concretized spaces. The entire framework is *topological* in and through. An example is the production of spacetime "points", a theme that was already developed in an earlier book [21], and was also a favourite thread in Whitehead's early mature philosophical work [12].

The role of topology in Russell space should not be underestimated because, at that time (1927), when the very concept of Hausdorff topological space was quite recent, it is noteworthy that Russell, a philosopher and mathematician, would suggest leaving behind the emerging field of Riemannian geometry, with the latter's traditional emphasis on the metric field, in order to propose what is, in a nutshell, a topological theory of fundamental physics based on an underlying event ontology of the world. Using some original topological methods from his own work but also Hausdroff's, Russell demonstrated how to technically construct any space you need out of a nexus of topologically interacting events. Two applications of this general method are a provisional attempt to integrate the Einstein–Eddington–Weyl concept of "dynamic spacetime" with the new quantum mechanics of Heisenberg, Schrodinger, Dirac, Born, and Jordan that was taking its final shape at the time of the writing of the book itself.[3]

Another application is the derivation of ordinary spacetime itself from the underlying events. This is the essence of Russell's (and Whitehead's) idea that in itself spacetime is not fundamental but should be derived from some more primordial level of nature, in his case the event structure of the world. In more recent times, the idea that spacetime is emergent have gained wide popularity, especially in the wake of John A. Wheeler's theory of "quantum foams" and similar concepts [45]. One may also recall David Bohm's speculative theories regarding the possible existence of a non-Lorentz invariant primordial random field-theoretic structural layer (sub-quantum fluctuations) underlying spacetime itself [46–48]. The idea of emergent spacetime was not invented by Russell. In fact, there is a strong evidence that the later Leibniz began to move into this direction in his mature philosophy, as can be sensed after reading some of his correspondence, especially with Arnauld [49].

But of course the theme of "emerging spacetime" has become quite popular in recent decades due to the increasing volume of researches conducted by some mainstream programs of quantum gravity such as noncommutative geometry, loop quantum gravity, causal net theories, string theory, Penrose spin foams and networks [3–5,50–54]. While the technical contents of each of these competing research diagrams differ significantly from each other, what somehow unifies most of them is the belief that quantizing gravity implies quantizing spacetime itself, and hence forces classical spacetime to become an emergent structure, while a kind of "quantum spacetime," e.g., spacetime governed by a quantized metric field operator, is more fundamental.

It is interesting to note that, technically speaking, Russell space can reconfigure itself to produce either discrete or continuous spaces. This reflects Russell's conscious awareness that a more fundamental underlying space could very well turn out to be discrete rather than continuous (basically due to quantization). The idea that spacetime might be ultimately discrete, say below the Planck length scale, is a popular subject nowadays. The most consistent proponents of this approach appear, at the moment, to be those working within loop quantum gravity, causal net theories, and few other research programs such as the group quantum field theory approach to gravity [55]. Regardless of which quantum gravity program will eventually succeed, I would like to conjecture that Russell space is rich, general, and complex enough to incorporate several of the main traits of the ultimate victorious theory (if any) destined to dominate the crowded battlefields of quantum gravity. The reason motivating my conjecture above is that Russell decided to deploy *ordinal* topological methods to implement reconfigurability in his generalized space

---

[3]   It should be noted that Russell did *not* use field quantization algorithms such as canonical quantization (the only one known by that time). Quantum field theory (QFT) as a general abstract subject was not really there yet.

concept. Being grounded in Cantorian set theory [10,24], Russell had no problem dealing directly with the *actual infinite* while invoking very strong principles such as the axiom of choice [56]. Therefore, his reconfigurable ontological framework of spacetime genesis, in my opinion, may be considered as relatively sophisticated enough to accommodate a wide variety of potential concrete theoretical proposals and general frameworks stemming from the ongoing research on quantum gravity, maybe by incorporating such inputs as "add-ons" to be appended into the ordinal nexus of the Russell space's event assemblage engendering the very production of spacetime as such.[4]

## 4. Additional Remarks on Russell and Quantum Gravity

However, before proceeding, I should provide some cautionary remarks here regarding the relation between Russell's ideas and *contemporary* theories of quantum gravity. The concept of what constitutes the essence of quantum gravity *as such* is itself historically evolving. When we put Russell's 1927 book in comparison with *present-day* research, the intention is not to suggest that somehow, magically, Russell had anticipated the technical content of today's theories of quantum gravity. As a matter of fact, three major key pillars of *contemporary* quantum gravity had not yet been known in 1927. These are:

1.　Quantum field theory (QFT).
2.　Microwave background radiation and cosmic expansion.
3.　Black hole thermodynamics.

QFT is at the heart of quantum gravity because gravitation is a *field* theory, so one needs to quantize a field right from the beginning. Cosmology and black hole radiation are essential for testing the physical content of a proposed quantum gravity theory [57]. Because of the lack of these components in Russell's approach to the subject, direct low-level technical comparison between 1927's and today's theories is not feasible. A high-level philosophical and formal (ontological) approach, such as the one we adopt here, might be more practical.

An example that might illustrate the need to be cautious when making historical comparisons is the last feature of Russell space quoted above, namely the reconfigurablity of this space to become either continuous or discrete. Now, in present-day formulations, loop quantum gravity and causal nets, just to mention two well-known examples, *derive* the discreteness of spacetime by solving something like a "quantum-gravitational eigenvalue problem" (computing the spectrum of a quantum field-theoretic operator), e.g., see [4,51,52]. This is the correct approach in the modern formulation, but did Russell do that? Hardly. Russell in fact was inspired by an *older* idea that predated modern physics itself. Riemann himself, in his major talk that launched differential geometry, had already speculated that fundamental spacetime in physics may very well turn out to be discrete [2,58]. To implement Riemann's insight, Russell used Cantorian methods, i.e., ordinal relation calculus [10,13–15,24,40], but with a novel "topologization" intersecting some early approaches to set-theoretic topology such as Hausdorff's [26]. In other words, this is a *philosophical* approach to the physics of nature. Russell did not bother to apply the canonical quantization procedure in his book,[5] preferring instead to operate with the full machinery of his ordinal topological space. That is a cautionary tale about the history and interpretation of the evolution of quantum gravity.

Nevertheless, Russell's theory (or set of theories and proposals) should *not* be dismissed as irrelevant because of the above historical issues. Quantum gravity is a large research program. What was understood by the term 'quantum gravity' in 1927, for example in Russell's mind, *has* evolved since then to our *current* conception of the field, which, in

---

[4]　In other words, it is possible to imagine that Russell space may be "upgraded" by updating it to handle new issues emerging later such as black hole thermodynamics, inflation, cosmic acceleration, etc. Clearly significant amount of technical work is needed, and, hence, the motivation behind this note, which is to encourage researchers to look into Russell's framework for inspiration and possible new technical ideas.

[5]　Pauli and Heisenberg's papers on quantum field theory were worked out shortly later, in 1928 and 1929 [9]. Modern QFT was effectively established only toward the end of the 1940s [59].

turn, may radically change in the near or far future. From the historical perspective, the year 1927, which saw the completion of quantum mechanics, also witnessed the publication of this first book-length study setting *both* quantum physics and spacetime physics against each other, searching for a possible harmony. Russell's attempt, of course, eventually did *not* succeed in this regard (fully unifying the quantum with the gravitational), but he deserves the historical credits of the first attempt (in a book form.)

## 5. Critical Remarks on Russell Space

Finally, I would like to provide some critical observations on Russell's concept of the event. As mentioned above, a core feature of Russell space is that it is a dynamic assemblage of fundamental basic building blocks called *events* (Whitehead's "blocks of becoming", a concept itself taken from Bergson). Those events are more real than points.[6] In any case, event assemblages and Russellian causality (the latter is not identical to conventional causality in mainstream philosophy of science and will not be discussed in detail below) are sufficient to derive the fundamental features of how the Russell space mechanism of producing "other emergent spaces" works in practice.[7] The production of spacetime is only one structural function implicit in Russell space. Other, more dynamic features (such as the incorporation of contextual fields) may also be absorbed into the Russellian scheme in future work in order to extend the scope of the ontological framework.

The idea that underlying nature is a deeper and, in a sense, more primordial level of the Real (an ontological layer that is comprised of event multiplicities) is older than Russell, dating back to at least Leibniz's monadology, which is a particularly famous example of monadic ontologies. The event-monad system was later invoked and used so brilliantly by Schelling in his profound 1800 book on the philosophy of nature [35]. Afterwards, the idea appeared with several thinkers such as Mach [31], James [32], Russell [21], Whitehead [12], and, most recently, the joint work of Deleuze and Guattari [63,64]. All these philosophies of nature share a certain commitment to what one might roughly call "monadological pluralism", an attitude that seems to be closely allied to one version of monism or another. As we learned from Gilbert Simondon, Deleuze, and Guattari, pluralism and monism are very closely related to each other: multiplicity implies the univocity of being, and, conversely, univocal being can be expressed in multiple ways [63–66].

However, it should be pointed out that Russell's thinking about the nature of the event was heavily influenced by two other figures: Albert Einstein and Alfred North Whitehead. The mutual relationships between the three thinkers is very complex, and a new detailed investigation of this subject must be left to other places. However, some specific technical details relating to the construction of the event as a concept are relevant to my overall aim in this article. First, note that Einstein's influence on Russell had always been filtered through the latter's relation with Eddington on one hand, and Weyl on the other. Indeed, these last two writers had shaped the *mathematical* theory of general relativity and gravitation, essentially altering all subsequent discussions of the topic, in fact more so than Reichenbach's impact on spacetime theories. Reichenbach was "more Russellian than Russell himself", but, ironically, Russell was not always as Russellian as he should had been.

In fact, one of those moments where Russell appears to have *faltered* is in the issue of modeling events as "blocks of spacetime", an idea advocated by Whitehead's Bergsonian interpretation of Einstein's gravitational theory [67]. In such a manner, Russell missed a great opportunity to *avoid* falling into the trap of *geometrizing* dynamics,[8] leading to the *pangeometrism* of contemporary mathematical physics, a problem that still haunts us

---

[6]  There is no complete, fully worked-out theory to be found in Russell's (nor in Whitehead's) accounts explaining how to derive the fundamental particles of physics from events, but hints and proposals on this subject matter abound in their texts, which stimulated a sizeable literature on interpretive strategies applied to Russell's and Whitehead's metaphysical systems, e.g., see [60–62].

[7]  For more insight into Russell's unique concept of causality, see how Reichenbach used temporal order to derive fundamental structures in general relativity and gravitation [44].

[8]  Reichenbach, but not Russell, succeeded in fully evading this geometric trap [44].

up to today. Temporality cannot be modeled as another dimension in a 4-dimensional manifold. A better approach, probably closer to the intuition of time in Bergson and Heidegger [68], might be to think of the event as a *topological flow* rather than an "eternal" or "frozen" block of becoming as in Whitehead [12]. It is Weyl's early stipulation that "Einstein and Minkowsky had effectively *dynamized* space by introducing time as a fourth dimension" [16] what caused such unfortunate chain of confusions about the relation between dynamics and spacetime. A Russell space free of the Russellian-Whiteheadian portrayal of events as "blocks of spacetime" would have been considerably more open to current problems of dynamics and entropic flows. Time probably cannot be spatialized as was originally conceived by the founders of relativity in the form of a fourth dimension even if this dimension is declared "time-like" [69].

## 6. Conclusions

I looked into Russell's 1927 masterpiece on the philosophy of nature, *The Analysis of Matter*, and suggested that this book, especially Part III, which is often neglected in the secondary literature, contains a concept of dynamic space, which I called Russell space, that may be considered to constitute a general framework for an abstract ordinal topological space that is quite broad in scope and notable for its ontological reconfigurability. Russell space may be used to understand the genesis or emergence of conventional spacetime as a case example, and can be configured for both continuous and discrete spaces. Moreover, despite not employing field quantization, it is suggested that, historically speaking, Russell's book may be viewed as the first book-length attempt to examine in philosophical terms the conceptual relation between quantum mechanics and gravitation.

**Funding:** This research received no external funding.

**Conflicts of Interest:** The author declares no conflict of interest.

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
