# Peer review of "On Russell’s 1927 Book The Analysis of Matter"

_philosophies, doi:10.3390/philosophies6020040_

Round 1

Reviewer 1 Report

This article is very well written, dense and interesting from a historical point of view. Nonetheless its main thesis: that Russell's Analysis of Matter offers a "comprehensive conceptual framework for quantum gravity" is more said than proven. But being such a short paper no more can be demanded and the statement of this thesis is in itself interesting enough.

Reviewer 2 Report

This is a short note connecting the third part of Bertrand Russell's Analysis of Matter with contemporary questions in the foundations of theoretical physics, specifically quantum gravitation. The heart of the argument is that in the little noticed, highly technical third section of the text, Russell develops a maximally general mathematical framework for space, Russell space. The history of mathematics through the end of the 19th, beginning of the 20th century saw mathematics moving from Euclidean space to general Riemannian space to even more general topological spaces. The argument in this work is that Russell, for epistemological reasons more related to physics than pure mathematics, was able to construct a more general notion of space and that it may be able to be used as the natural home of the quantum gravitational field that contemporary physicists are seeking to describe because it would be discrete instead of the usual continuous space. This is because the physicists, and thereby the epistemologists, demand that events be the basic ontological building block.

I will admit that on the face of it, the thesis seemed absurd at first glance.  Russell's work was being done in the mid-20's when the sort of field theories we now consider were first being envisioned by Hermann Weyl and then Albert Einstein. How could Russell have internalized the idea so quickly and then expanded upon it in such a remarkable fashion? But the author is a careful expositor of Russell and demonstrates a firm grasp of both the mathematical tools and the historical development of the early 20th century physics. Further, the case is made provisionally, not overstating the result and the piece leaves promissory notes in crucial places that would require technical work to be done. What is here is not a finished project, but rather a sort of proposal that makes the case that the broader project is important and likely to be the case. The proposal is convincing.

And that, of course, is the weak point of the paper. It puts the taste in your mouth of a significant result and gives you reason to think it will be delivered to the table, but it is just the menu -- not the dish. There are likely at least three full length papers that will come out of this note before the author is done. A careful and complete discussion of part III of the Analysis of Matter is needed. A clear explication of the concept of Russell space and its relation to Hausdorff space and its other mathematical progenitors as well as the foundations of emergent spacetime in Leibniz are needed (that might be two papers). And then at the end of the paper in two tantalizing paragraphs the author teases a comparative discussion of Russell and Reichenbach. That is a paper unto itself. This could be a series of papers or a complete manuscript. It would be a book worth working through.

But all we have is the introduction, the prolegomena to this project.  But for now, it will have to suffice. I strongly favor publication of it with one small change. On line 119, the phrase "Russell is no ordinary epidemiologist" most likely was meant to read "Russell is no ordinary epistemologist" (unless he had a medical side I know nothing about).

Reviewer 3 Report

Please read the attached file.

Round 2

Reviewer 3 Report

I thank the Author for the substantial improvements he made to the original manuscript before all in its too pretentious original title. Moreover, the adding of the new Section 4 located the contribution of Russell in a more adequate place with respect to the further developments of QG and QFT. In this framework,  the suggestion to scholars, aim of the note, to study this original contribution of Russell to the mathematics of fundamental physics present in this book is fully justified and appreciable.